# The Validity and Reliability of Two Commercially Available Load Sensors for Clinical Strength Assessment

**DOI:** 10.3390/s21248399

**Published:** 2021-12-16

**Authors:** Kohle Merry, Christopher Napier, Vivian Chung, Brett C. Hannigan, Megan MacPherson, Carlo Menon, Alex Scott

**Affiliations:** 1Department of Physical Therapy, University of British Columbia, Vancouver, BC V6T 1Z3, Canada; chris.napier@ubc.ca (C.N.); alex.scott@ubc.ca (A.S.); 2Centre for Hip Health and Mobility, Vancouver, BC V5Z 1M9, Canada; vivian.chung@hiphealth.ca; 3Menrva Research Group, School of Mechatronic Systems Engineering, Simon Fraser University, 250-13450 102 Avenue, Surrey, BC V3T 0A3, Canada; brett.hannigan@hest.ethz.ch (B.C.H.); carlo.menon@hest.ethz.ch (C.M.); 4Department of Health and Technology, ETH Zürich, Lengghalde 5, 8008 Zürich, Switzerland; 5School of Health and Exercise Sciences, University of British Columbia, Kelowna, BC V1V 1V7, Canada; megan.macpherson@ubc.ca

**Keywords:** evaluation studies, hand-held dynamometry, muscle testing, sensor characterization, validity and reliability check

## Abstract

Objective: Handheld dynamometers are common tools for assessing/monitoring muscular strength and endurance. Health/fitness Bluetooth load sensors may provide a cost-effective alternative; however, research is needed to evaluate the validity and reliability of such devices. This study assessed the validity and reliability of two commercially available Bluetooth load sensors (Activ5 by Activbody and Progressor by Tindeq). Methods: Four tests were conducted on each device: stepped loading, stress relaxation, simulated exercise, and hysteresis. Each test type was repeated three times using the Instron ElectroPuls mechanical testing device (a gold-standard system). Test–retest reliability was assessed through intraclass correlations. Agreement with the gold standard was assessed with Pearson’s correlation, interclass correlation, and Lin’s concordance correlation. Results: The Activ5 and Progressor had excellent test–retest reliability across all four tests (ICC(3,1) ≥ 0.999, all *p* ≤ 0.001). Agreement with the gold standard was excellent for both the Activ5 (ρ ≥ 0.998, ICC(3,1) ≥ 0.971, ρ_c_ ≥ 0.971, all *p*’s ≤ 0.001) and Progressor (ρ ≥ 0.999, ICC(3,1) ≥ 0.999, ρ_c_ ≥ 0.999, all *p*’s ≤ 0.001). Measurement error increased for both devices as applied load increased. Conclusion: Excellent test–retest reliability was found, suggesting that both devices can be used in a clinical setting to measure patient progress over time; however, the Activ5 consistently had poorer agreement with the gold standard (particularly at higher loads).

## 1. Introduction

Due to aging, injury, or a combination of the two, many people experience strength deficits at some point in their lifetime [1,2]. Strength deficits are apparent in a variety of conditions including osteoarthritis [3], stroke [4], and tendinopathy [5], to name a few. Such deficits can result in mobility limitations [6,7] and may predict mortality [8], length of hospital stay [9], and hospital readmission rates [10]. Conservative management strategies (e.g., therapeutic exercise targeting mechanical tissue deficits) can lead to positive tissue adaptation and symptom alleviation in a variety of conditions [11,12]. For example, conservative Achilles tendinopathy treatments, including strengthening exercises, are recommended as a first step prior to surgical intervention [13], and resolve approximately 75% of all cases [14]. Given the evidence for strength-based exercises, muscle strength assessment is an important component within physiotherapy practice [15]. Clinicians use muscle strength evaluation to benchmark the level of impairment and guide subsequent treatment and return to activity [16]. It is therefore imperative that clinicians be able to assess and monitor muscle strength throughout the course of rehabilitation for accurate exercise prescriptions. 

Several methods for evaluating muscle strength are commonly used in a clinical setting, including manual muscle testing (MMT), handheld dynamometry (HHD), and isokinetic dynamometry (ID), among others. MMT is a subjective measure of muscle strength whereby a clinician evaluates the amount of force a patient can exert or withstand on a given extremity. HHD is a portable, objective measurement tool which can be used to quantify muscle strength across the upper and lower extremities. HHDs typically have a range of approximately 0 to 150 kg and can cost as low as $200 USD, but commonly exceed $1000 USD [17,18]. ID is another objective measurement tool which is generally fixed in position and can be used to characterize muscle strength and limb kinetics (e.g., joint moments). IDs typically have a range of approximately 0 to 600 Nm and often start at $10,000 USD [18]. MMT is ubiquitous within physical therapy practice, though it has demonstrated poorer results for assessing muscle strength when compared to objective measures such as HHD and ID [16,19,20]. Despite ID being highly regarded for assessing muscle strength [18], HHD and ID are highly correlated in certain measurement conditions such as lower extremity peak torque assessment [21], measurements of quadriceps strength [22], and upper extremity muscle strength assessment [18]. While HHD presents several advantages when compared to ID in terms of portability, ease-of-use, and testing time [16,18,23,24], its cost still poses a burden for applications in real-world settings as it may not allow clinicians to engage in continued home-based monitoring of exercise over time, highlighting a need for low-cost consumer products. Further, the loading range and accuracies for both HHDs and IDs may exceed what is needed for clinical strength assessments [25] which may allow clinicians to use more compact and cost-efficient devices.

The Activbody Activ5^®^ load sensor (Activbody Inc., San Diego, CA, USA) and the Tindeq Progressor^®^ load sensor (Tindeq, Trondheim, Norway) are two accessibly priced ($149.99 [26] and $135.00 [27] USD, respectively) and commercially available fitness devices which incorporate biofeedback. The Activ5 uses a Bluetooth-enabled compressive load sensor to measure muscle strength which is then displayed to the user via an associated smartphone application. The ergonomic design allows the Activ5 to be used in several positions targeting various body parts and muscle groups, and the smartphone application enables individualized exercise programming. The Progressor uses a Bluetooth-enabled tensile load sensor to measure muscular endurance, peak force developed, or the rate of force development. While initially developed for rock and mountain climbers, the attachment points allow the Progressor to be easily adapted for nonclimbing exercise training.

To the authors’ knowledge, no previous studies have assessed the validity and reliability of the Activ5 or the Progressor for force measurement. For the clinical application of such commercial technologies, clinicians and patients must be cognizant of the device’s strengths and limitations for measuring health analytics prior to prescribing or using them within a rehabilitation intervention. One way to assess measurement capabilities of load sensors is by performing a bench study in a mechanical testing machine, comparing the devices with a gold standard to assess the accuracy and replicability of force measurements obtained with each device. 

The aim of this study was to assess the validity and reliability of the Activbody Activ5 and Tindeq Progressor under controlled conditions. Both devices were directly compared against a gold-standard test instrument. We hypothesized that both devices would (a) demonstrate high test–retest reliability, and (b) display strong correlations with the gold standard over the range of interest, with agreement deteriorating as loading increased towards the maximum design load of each device.

## 2. Materials & Methods

Both the Activbody Activ5 and the Tindeq Progressor load sensors were mechanically tested to assess measurement validity and reliability. Technical specifications of the Activ5 and Progressor can be found in Table 1. Both load sensors were tested in an Instron ElectroPuls E10000 universal testing machine (UTM) (Instron, Norwood, MA, USA) using the Instron Dynacell 2527-202 load cell (10 kN capacity). The Instron ElectroPuls E10000 was considered as the gold-standard measure for this protocol. The force measurement error of this UTM is stated by the manufacturer to be within ±0.5% of applied load, or ±0.5 N, whichever is greater. Further, calibration of device linear displacement accuracy noted a maximum uncertainty of ±0.015 mm. Activbody does not publish the Activ5’s calibration metrics. The Progressor is calibrated by Tindeq at 0 and 63 kg and assumes 100% linearity over its loading range.

### 2.1. Experimental Protocol/Procedures

Data collection was conducted over a single testing day. The experimental setup for testing of both load sensors can be found in Figure 1. Devices were fixed in the UTM differently during testing due to the Activ5 being compression-based and the Progressor being tension-based. From a side profile, the Activ5 device is roughly teardrop shaped thereby creating a nonuniform bearing surface when placed on a flat surface. To avoid point-loading and potentially cracking the Activ5’s plastic shell during testing, a thermoplastic interface comprised of ethylene-vinyl acetate and hydrocarbon resin was molded to conform to the upper and lower surface of the Activ5 prior to testing. The side of the thermoplastic interface not touching the Activ5 was flattened to provide a flat bearing surface for compressive testing in the UTM. Specifically, after heating the thermoplastic to a semisolid state, the thermoplastic was poured on to one side of the Activ5. Following this, a flat piece of aluminum was centered and temporarily clamped above the Activ5 with sufficient pressure to extrude the thermoplastic towards but not beyond the edge of the Activ5 shell. The applied pressure was within the maximum design load of the Activ5. After fully cooling, the same process was repeated with the opposite side of the Activ5. This interface simulates a more biofidelic representation of actual device usage when contrast with point pressure, as it mimics the palms of a person which is typical of the suggested use of the device. The thermoplastic interface was allowed to cure completely prior to testing and was checked between tests for signs of wear or deformation. All components besides the Activ5 and the thermoplastic interfaces along the loading axis were made of metal. The Progressor has two fixed circular attachment points from which to load the device in tension. The Progressor was fixed to the UTM base plate with a screw pin anchor shackle, and to the UTM load cell on the upper surface with a carabiner, both made of stainless steel. All components along the loading axis (including the Progressor) were made of metal.

A single Activ5 and Progressor unit were used during all testing. All tests were completed at 23 °C and were completed 10 min apart. Both devices use replaceable batteries which were replaced with fresh batteries prior to testing. After each test, devices were disconnected/reconnected to their respective smartphone applications and were zeroed using the tare feature. The UTM load cell sampled at 100 Hz and only applied a linear displacement. Data were collected for the Activ5 through a custom script running on a Bluetooth-enabled computer sampling at 10 Hz. Data were collected for the Progressor by screen-recording a live data view of the Progressor’s smartphone application; the Progressor sampled at 5 Hz at the time of testing. Four different test types were used to assess both devices across the entire loading range while also mimicking exercise-based use cases (Figure 2). 

A stepped loading test was conducted testing each load sensor over the entire loading range. During the stepped loading test, devices were subjected to a series of test forces selected based on ASTM E4-16: Standard Practices for Force Verification of Testing Machines. The Activ5 was tested at 10 different forces: 10, 20, 40, 70, 100, 200, 400, 550, 700, and 883 N. The Progressor was tested at 11 different forces: 10, 20, 40, 70, 100, 200, 400, 700, 1000, 1236, and 1471 N. Each load was held for 5 s and was separated from adjacent loads by a 5 s ramp before a final 5 s unload.

A stress relaxation test was conducted to assess any stress relaxation affects which may take place over a short time period; stress relaxation describes a reduction in stress while strain is held constant. A short-duration stress relaxation test was selected, as this timeframe is akin to a medium-duration isometric exercise. During the test, a 10 s ramp was applied before a 14 s hold at approximately maximum design load (883 N for Activ5, 1471 N for Progressor) followed by a 3 s unload. 

A simulated exercise test was conducted to assess load sensor agreement during a simulated isometric exercise in which a rapid ramp is followed by an isometric hold and rapid unload. This test was designed to mimic an individual conducting an isometric exercise repetition in which loading onset is quick, followed by a static hold, and expedited unload. During the test, a 2 s ramp was applied to each device to achieve a force of approximately 50% maximum design load (440 N for Activ5, 700 N for Progressor) which was held for 14 s before a 3 s unload. 

Lastly, a hysteresis test was conducted to assess load sensor hysteresis error. Hysteresis describes any measurement deviation in response to increasing to a specified load from zero when compared to the same load achieved by decreasing from maximum load. During the test, a 30 s ramp was applied taking both load sensors to approximately maximum design load, followed immediately by a 30 s unloading period.

### 2.2. Data Analysis

Data files corresponding to all loading variables were exported from the UTM in csv format through the WaveMatrix™2 testing software (Instron, Norwood, MA, USA). Data files from the Activ5 were exported from the custom computer script in csv format. Data files from the Progressor were transcribed into csv format from the smartphone application screen recordings using a free video editing software (VDSC Free Video Editor, Multilab LLC, Spring Lake, NJ, USA). Data resampling was completed using a custom MATLAB script (The Mathworks; Natick, MA, USA). Resampling was completed to match the UTM sampling rate to that of the Activ5 (10 Hz) and Progressor (5 Hz). From the 100 Hz UTM data, data points were selected which most closely matched the timestamp of the lower resolution load sensor data. For example, if an Activ5 data point corresponded to 5.0 s, the UTM data point most closely matching 5.0 s was selected from the 100 Hz data and extracted as the resampled data point. 

Once UTM data were resampled to match that of the Activ5 and Progressor, the three repetitions of each test type were time synchronized with one another. Because both the Activ5 and Progressor data capture methods (custom script and smartphone application, respectively) required manually starting data acquisition, a slight time offset was present between the three repetitions despite exposure to the same preprogrammed UTM loading profiles. To correct this offset, two of the three data sets were manually shifted forward or backward in time based on distinguishable characteristics of the loading profiles. Time synchronization was not needed for the UTM data. Once all three repetitions for a given test type were synchronized in time with respect to one another, the three repetitions were averaged resulting in a single data set for each test type for each device (UTM, Activ5, and Progressor). A further time synchronization step was then conducted to synchronize the averaged UTM data set with the corresponding averaged Activ5 or Progressor data sets. Data were again manually shifted forward or backward with respect to time based on the same distinguishable characteristics of the loading profiles.

The stepped loading test was further analyzed at each of the test loads applied (10 for the Activ5, 11 for the Progressor). At each of these load levels, 5 s of data were analyzed starting when the UTM began ‘holding’ at a specific test load. Stress relaxation effects were analyzed over the 14 s holding duration during the stress relaxation tests. During this hold time, the position of the UTM was held constant after reaching the test load. Load sensor hysteresis effects were analyzed during the hysteresis tests by taking the difference between two force measurements for the same load, one captured by increasing force from 0 to a specific load, and the other captured by decreasing force from the maximum load to the specific load of interest.

### 2.3. Statistical Analysis

Scatterplots of all individual tests were checked for outliers. Test–retest reliability was analyzed using the intraclass correlation coefficient (ICC) to assess repeatability. In accordance with Koo and Li’s [28] suggested standards for reporting ICC estimates, SPSS statistical package version 27 (SPSS Inc., Chicago, IL, USA) was used to calculate ICC estimates and their 95% confidence intervals based on a two-way mixed effects model, using single measures, and absolute agreement. Further, reliability associations were classified based on the 95% confidence intervals of the ICC estimate as poor (0–0.49), moderate (0.50–0.74), good (0.75–0.89), or excellent (0.90–1.00) [28]. Quantitative agreement between the gold-standard UTM and both load sensors was assessed using Pearson’s correlation (demonstrating precision), interclass correlations, and Lin’s concordance correlation (demonstrating systematic bias) [29,30]. Values of Pearson’s correlation coefficient and Lin’s concordance coefficient were classified as little or no relationship (0.00–0.24), fair relationship (0.25–0.49), moderate-to-good relationship (0.50–0.74), or a good-to-excellent relationship (above 0.75) [31]. A *p* value was considered statistically significant if <0.05.

## 3. Results

Data collection errors resulted in several trials ending prematurely. Specifically, the UTM data files for two repetitions of the stress relaxation test and two repetitions of the simulated exercise test comparing the Progressor to the UTM ended prematurely and not all unloading data were captured (Figure 2F,G). To ensure the resulting test averages consisted of data from all repetitions of a given test type, data sets corresponding to both tests were reduced to the duration of the shortest test. Additionally, two outliers were detected by visual inspection of the scatterplots. One data point was identified in one repetition of the stepped loading test for the Progressor, and one data point was identified in one repetition of the hysteresis test for the Progressor. In both cases, data dropped from 996 N (stepped loading) and 551 N (hysteresis) to 12.7 N and 12.4 N, respectively, before increasing to expected levels in the next data points (996 and 581.2 N, respectively). Note that the time between each data point is 200 milliseconds. To correct the data, linear interpolation was conducted based on adjacent data points.

Excellent repeatability was noted for both devices across the three repetitions for each of the four test types (Table 2). When compared to the UTM, both devices demonstrated excellent reliability (ICC) and relationships (Pearson’s and Lin’s concordance), indicating strong reproducibility when comparing measured to applied force (Table 3).

The input–output relationship of both devices was obtained by plotting the mean force measured across the three testing repetitions by the specific device (Activ5 or Progressor) against the mean applied force applied by the UTM (Figure 3). The Activ5 underreported force over the entire loading range, with deviation from applied force increasing at higher loads; this is consistent with the stated error of the device (Table 1) which increases as a function of the applied load. In general, the Progressor’s input–output response was almost linear with little error noted over the entire loading range. Deviations were noted for the Progressor during the simulated exercise test, particularly during the rapid ramp from an unloaded state to the test load.

Accuracy over the entire loading range was highlighted by observing measured forces for each device across a discrete number of test loads during the stepped loading tests and comparing those to forces applied by the UTM (Figure 4). Both devices underreported the force applied by the UTM at all test loads. The measurement error for both devices also tended to increase towards higher loads, though the specific amount of error varied substantially between the two devices. At low loads (10 to 100 N), the Activ5 reported errors less than 6.2 N, whereas at higher loads (550 to 883 N) error increased from 61.6 to 149.7 N, respectively. While staying within the Activ5’s manufacturer reported accuracy at the low loads, error was approximately double what would be expected at the 550 N test load (33.7 N expected, 61.6 N recorded) and triple what would be expected at the 883 N test load (50.4 N expected, 149.7 N recorded). In contrast, at low loads (10 to 100 N), the Progressor reported errors less than 0.9 N, whereas at higher loads (1000 to 1471 N) errors increased from 5.1 to 7.3 N, respectively. The Progressor’s level of error falls within the margin of error of the UTM, indicating excellent accuracy across the entire loading range.

Stress relaxation effects resulted in a change in applied force of 16.3 N for the Activ5 (corresponding to a 25.9 N change for the UTM) over the 14 s hold when displacement was held constant (Figure 5A). For the Progressor, stress relaxation effects resulted in a change in applied force of 0.2 N (corresponding to a 0.7 N change for the UTM) over the 14 s hold when displacement was held constant (Figure 5B).

Maximum hysteresis, calculated as the maximum difference in the measured force for a given applied load in the loading and unloading directions, was found to be 35.0 N (4.9% of maximum force difference during hysteresis test) for the Activ5, and 33.5 N (2.3%) for the Progressor. Average hysteresis over the entire hysteresis test cycle was 21.2 N for the Activ5 (3.0% of maximum force difference) and 10.4 N (0.7%) for the Progressor.

## 4. Discussion

This work assessed the validity and reliability of two commercially available Bluetooth load sensors (Activbody Activ5 and Tindeq Progressor) under controlled conditions. The test–retest reliability of each device was assessed by comparing a series of different tests designed to investigate the full loading range over different durations. Additionally, each device was compared to a gold-standard UTM to assess agreement. 

### 4.1. Primary Findings

Both devices demonstrated excellent test–retest reliability across all test types. Measurement repeatability is critical for rehabilitation and health-related applications to ensure that measurements are consistent over time, to allow for evaluation of patient progress. Additionally, both devices demonstrated excellent agreement compared to the UTM across all test types. Despite this excellent agreement, the Progressor outperformed the Activ5 in all testing scenarios by better aligning with the UTM, as indicated by the higher agreement statistics. By isolating various test loads over the entire loading range of each device, we were able to comment on the validity of both systems relative to the UTM.

The Activ5’s accuracy is relatively low at loads below 40 N, and deteriorated substantially at loads above approximately 400 N, or roughly 50% of its maximum design load. The range between 40 to 400 N (where the Activ5’s accuracy is notably higher) may overlap with loads typical of upper-body isometric exercises. For example, a fifth percentile male with elbows at varying degrees of flexion (from 60° to 180°) generates a combined compressive force with both hands of approximately 145 N to 190 N [32]. Further, Bohannon (1997) [25] tested the maximal voluntary contractions of various muscle actions for males and females between the ages of 20 and 79 using a HHD [25]. Across the upper and lower body, forces ranged from approximately 100 to 400 N with only knee extension crossing 400 N consistently for both males and females. These values generally fall within the range for the Activ5 where accuracy is highest. Coupled with the high test–retest reliability identified in this study, the Activ5 provides users with a repeatable measure, with error varying by approximately 5.1 N to 36.5 N over the 40 to 400 N range. This error is considerable when compared to common HHD units such as the Lafayette Hand-Held Dynamometer or the MicroFET^®^3 Dynamometer, which have accuracies of ±1% of the force reading. However, because Activ5 users would typically be using the device over multiple days and comparing results with their personal history tracked on the associated smartphone application, this measurement error may rarely be an issue for the common user. Overall, the Activ5 showed excellent test–retest reliability; however, users should consider the tolerance for error to determine if the Activ5 is suitable for a specific use case.

In contrast to the Activ5, the Progressor demonstrated excellent accuracy over its entire loading range (which is 66% greater than the Activ5’s loading range). The Progressor’s error consistently increased as a function of applied load but stayed aligned with the UTM’s margin of error. Like the Activ5, most intended users of the Progressor likely use the device as a gauge of personal output or development over time. Because of the excellent test–retest reliability and validity demonstrated by the Progressor over its entire loading range, users may choose to use the Progressor for a variety of use cases.

### 4.2. Stress Relaxation Test

Stress relaxation, a phenomenon primarily attributed to viscoelastic materials, is associated with a time-dependent stress response under constant strain and primarily affects polymeric materials (though it can be found in metals subjected to sufficiently high temperatures) [33]. Metals typically exhibit a smaller stress relaxation response due to their high stiffness compared to polymers. Additionally, environmental characteristics (e.g., temperature) and the specific loading regime (e.g., loading speed, magnitude, duration) can influence stress relaxation effects [33]. In the context of this study, a stress relaxation test was performed with a short hold time to simulate a medium-duration isometric exercise. 

When the UTM was held at a constant displacement after loading both devices to their respective maximum design loads, a stress relaxation response was noted in the Activ5, but not the Progressor. Several factors could have contributed to this response. The Activ5 shell is made of a hard polymer whereas the Progressor is made of metal along the axis of loading, giving it a higher resilience to resist such effects. Additionally, the load magnitude could have caused the materials to leave the linear elastic loading range and exhibit a different stress relaxation response due to small levels of plastic deformation [34], though sufficient stresses were likely not reached as polymer yield strengths typically exceed 25 MPa with metals far exceeding that [33]. Lastly, the duration of the hold (14 s) may have contributed, though it would be expected that the stress relaxation response would increase with time, resulting in even lower forces measured if the test duration was increased. As to the extent and implications of the stress relaxation effects, a 16.4 N change in force was measured by the Activ5 over the 14 s hold. Depending on the intended use, this change may be negligible when compared to the 149.7 N error identified between the UTM and the Activ5 for the specific test load. Nonetheless, potential users of the Activ5 should be aware of this effect, and further testing to comprehensively detail the Activ5’s stress relaxation response is required.

### 4.3. Load Sensor Hysteresis

Load sensor hysteresis effects contribute to the combined error of a load sensor, thus influencing device utility. Maximum hysteresis effects were similar for both devices (35.0 N for Activ5 and 33.5 N for Progressor), though when normalized for maximum force difference, the maximum hysteresis effects were more pronounced in the Activ5 (4.9% for Activ5; 2.3% for Progressor). Averaging over the entire hysteresis test further reinforced the larger hysteresis effects in the Activ5 (average hysteresis of 21.2 N, 3.0% of max force difference) in contrast to the Progressor (10.4 N, 0.7%). Several factors may have contributed to these responses including material properties, device geometry, temperature, loading rate, and the device’s loading history [35,36]. The sampling frequency of both devices may have also contributed to the noted hysteresis effects. A slower loading rate (or a faster sampling rate) would increase the resolution of the devices and may decrease variation caused by relatively large increases in load. For example, by increasing the Progressor from an unloaded state to maximum design load (1471 N) at a fixed duration of 30 s, the ramp rate is equal to 49.1 N/second. Since the Progressor was measuring at 5 Hz, this resulted in a change of approximately 9.8 N per data point. As a result, slight deviations in measurement were likely magnified due to the rapid rate of change. A similar situation can be witnessed in the simulated exercise test during the rapid ramp from an unloaded state to approximately 50% of maximum design load (Figure 3G). Notable deviations in the input/output relationship are likely attributable to a ramp rate of 350 N/second, where the Progressor struggles to ‘keep up’ with the loading rate applied by the UTM. Future works seeking to use this result should verify hysteresis effects using different loading rates.

### 4.4. Use as a Clinical Measurement Tool

Despite some functional similarities to HHDs, the Activ5 and Progressor were not developed as clinical measurement tools; therefore, their use case in a clinical setting or as a home-based monitoring tool must be prefaced with several caveats. First, when comparing values with previously identified reference values (e.g., Bohannon et al. [25]), it is of the authors’ opinion that clinicians use the Progressor, an evidence-backed HHD (e.g., Lafayette Hand-Held Dynamometer [22,24,37], MicroFET3 Dynamometer [38,39]), or ID (e.g., Biodex [22,40]). Further, clinicians must consider the movement type they wish to assess before deciding which tool to use, as the Progressor is a tension-based system while most HHDs come with attachments for compression or tension testing. Based on this study, the Activ5 should be used with caution when attempting to measure forces accurately for comparison to previously identified reference values; however, the Activ5 may be useful in assessing individual changes over time through repeated Activ5 measurements. Second, prior to using either device, users should be aware of their tolerance for error and whether they intend to use the same device over time. Both devices had excellent test–retest reliability, and in cases where the patient would be compared against themselves, the effects of inaccuracy would be nullified. In contrast, if a clinician uses multiple devices to measure muscle strength at several appointments (e.g., HHD at one visit, Activ5/Progressor at another visit due to device availability in the clinic), the inaccuracy of the Activ5 may result in inconsistent measurements and inaccurate evaluation of a patient’s muscle strength over time. However, this would also depend on the muscle groups and activities being tested as these significantly change the range of forces being applied [25]. If a clinician is testing a muscle group/activity with a smaller loading range (e.g., shoulder lateral rotation force or wrist extension force), the strength progressions would be smaller and therefore the Activ5 may be an inappropriate tool due to error potentially masking progression. Depending on the intended application, clinicians may be able to tolerate the Activ5’s error, particularly in the 40 to 400 N range where error is lowest, though consideration should be given as to what is considered tolerable in the context of prescription or revision of an exercise therapy program. Finally, prior to either device being classified as a “medical device”, the Progressor and Activ5 would need to undergo additional testing and validation with clinical populations and follow governmental regulations necessary to be identified as medical devices. For example, prior to being sold on the Canadian market, healthcare-related devices require a license under the Food and Drug Act.

### 4.5. Limitations

While we are confident that both devices had a conservative loading history and were in good working order, only a single Activbody Activ5 and a single Tindeq Progressor device were used during testing. Future research should assess interunit variability for both load sensors. Additionally, because these devices are designed primarily for commercial use, consistent software upgrades may invalidate some results as the device firmware and associated smartphone applications are modified. The Activ5 data collection was completed via a custom script running on a computer (not on the native smartphone application), therefore future device firmware upgrades may impact results. The Progressor’s data was collected via the built-in screen-recording feature on a Samsung smartphone at a frame rate which exceeded the Progressor’s sampling frequency at the time (5 Hz). The Progressor has since been updated to record at 80 Hz; therefore, some results identified in this work may improve (e.g., the effects of loading rate) though this change would only improve the Progressor’s performance.

Another limitation concerns the Activ5 experimental setup. The thermoplastic interface located on the upper and lower surfaces of the device during loading was necessary to promote even load distribution over the device’s surface (similar to normal Activ5 use), reduce the risk of cracking the device shell, and reduce a larger creep response from the shell of the device. Additionally, the response of both devices across different percentages of battery charge was not tested. This interface could have influenced the results, particularly those of the stress relaxation test. Further limitations associated with the number of testing repetitions, the uniform testing temperature, and the little variation in loading rate prompt the need for future works to confirm and expand upon the results presented.

## 5. Conclusions

This study assessed the validity and reliability of two commercially available Bluetooth load sensors, the Activbody Activ5 and the Tindeq Progressor, for the purpose of assessing muscle strength clinically. Our findings indicate excellent test–retest reliability for both devices, as well as excellent agreement with a gold-standard UTM, suggesting that both devices may serve as a cost-effective alternative for measuring force-related health analytics such as muscle strength. While the Progressor’s accuracy was comparable to the UTM over the Progressor’s entire force range, the Activ5’s accuracy substantially decreased at loads over 50% of the device’s force range, culminating in an error roughly 3 times that reported by the manufacturer at the maximum design load. The Progressor performed better during all test types, though it is challenging to draw a direct comparison for potential users as the Activ5 is a compression-based system while the Progressor is tension-based. Further investigation pertaining to interdevice variability and the effects of loading rate and temperature are warranted.

## Figures and Tables

**Figure 1 sensors-21-08399-f001:**
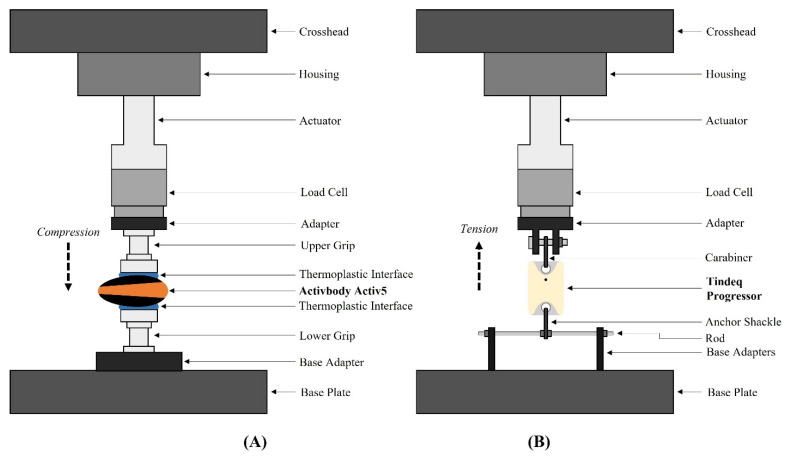
Schematic of the testing set-up using the universal testing machine for loading the (**A**) ActivBody Activ5 in compression; and (**B**) Tindeq Progressor in tension.

**Figure 2 sensors-21-08399-f002:**
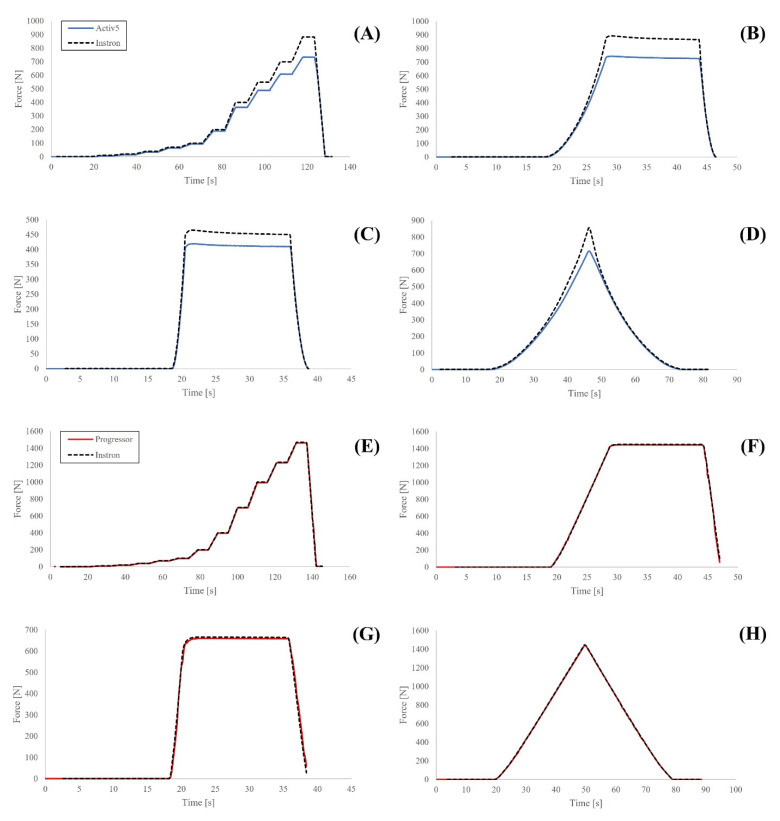
Loading profiles of the ActivBody Activ5 compared to the gold standard (Instron) for (**A**) Stepped Load Test; (**B**) Stress Relaxation Test; (**C**) Simulated Exercise Test; and (**D**) Hysteresis Test. Additionally, loading profiles of the Tindeq Progressor compared to the gold standard (Instron) for (**E**) Stepped Load Test; (**F**) Stress Relaxation Test; (**G**) Simulated Exercise Test; and (**H**) Hysteresis Test.

**Figure 3 sensors-21-08399-f003:**
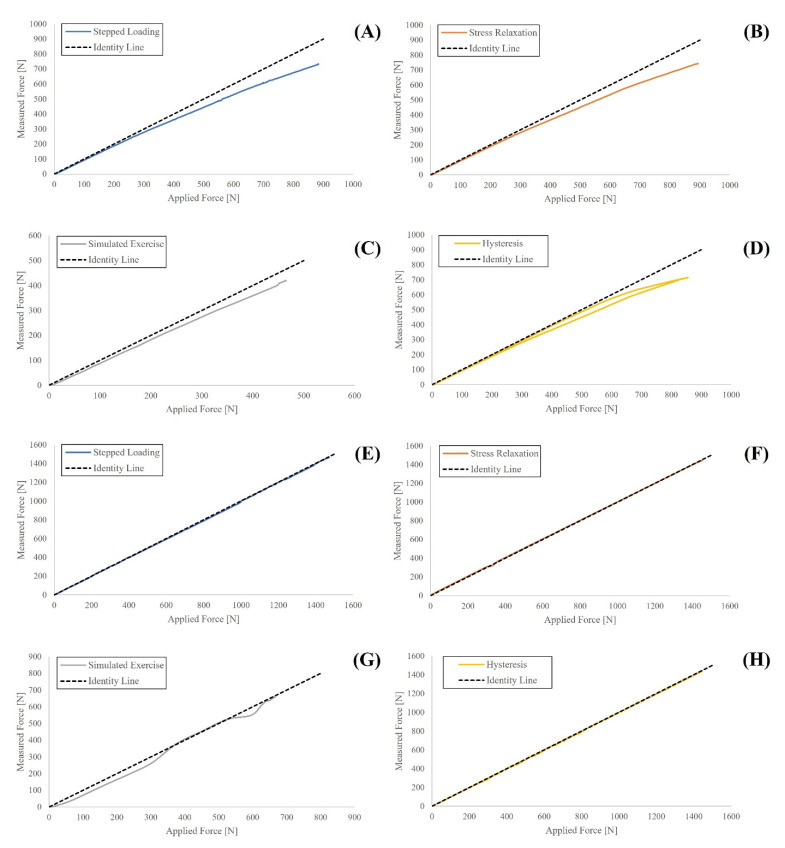
Mean force values across three testing repetitions of each test type comparing the force applied by the gold standard (Instron) against the force measured by the ActivBody Activ5 for (**A**) Stepped Load Test; (**B**) Stress Relaxation Test; (**C**) Simulated Exercise Test; and (**D**) Hysteresis Test. Additionally, force applied by the gold standard (Instron) against the force measured by the Tindeq Progressor for (**E**) Stepped Load Test; (**F**) Stress Relaxation Test; (**G**) Simulated Exercise Test; and (**H**) Hysteresis Test. The identity line indicating a perfect linear relationship is also shown.

**Figure 4 sensors-21-08399-f004:**
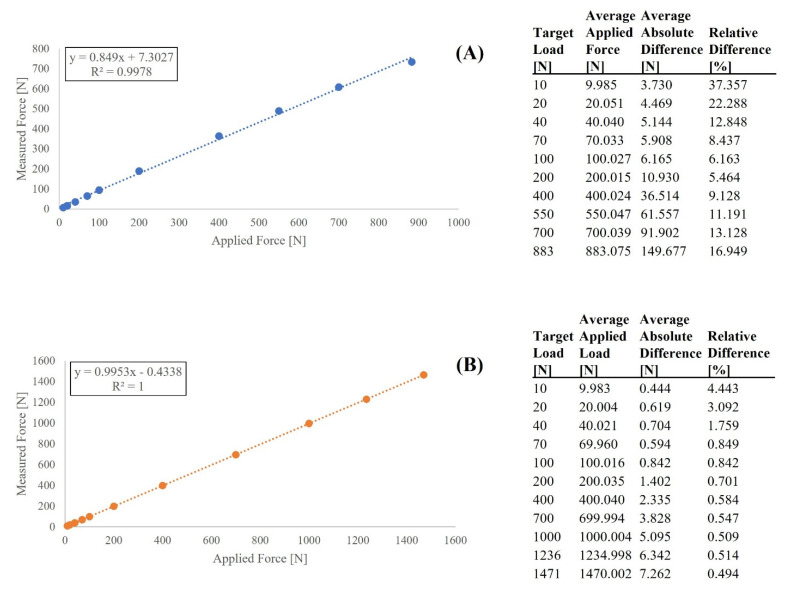
Relation between the force measured by each device and the force applied by gold standard (Instron) for the stepped loadings test. (**A**) The ActivBody Activ5 was tested to 10 different loads over the entire loading range; (**B**) The Tindeq Progressor was tested at 11 different loads. Reported data at each force level correspond to 5 s of measurement when force plateaued at each test load. For both the absolute and relative difference columns, values denote applied force minus measured force. The best-fitting line is also displayed for reference.

**Figure 5 sensors-21-08399-f005:**
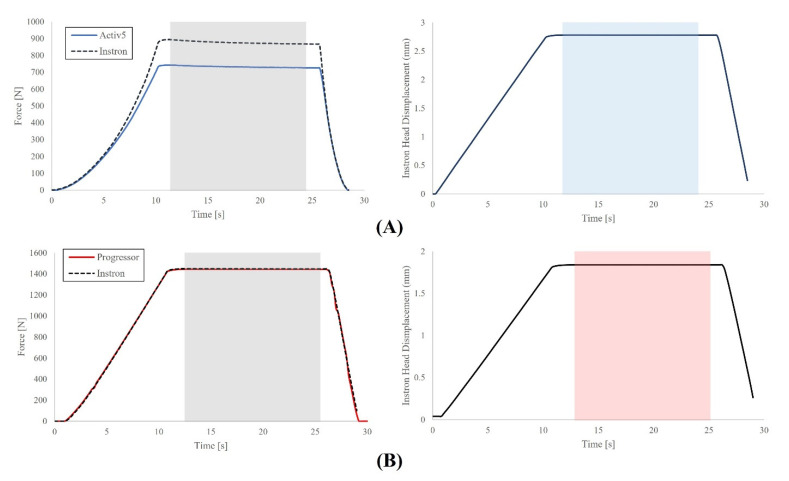
Stress relaxation effects for (**A**) ActivBody Activ5; and (**B**) Tindeq Progressor. Transparent regions denote isolated 14 s hold.

**Table 1 sensors-21-08399-t001:** Specification of consumer-grade load sensors.

	ActivBody Activ5	Tindeq Progressor
Weight [kg]	0.131	0.150
Dimensions (L × W × H) [mm]	95 × 78 × 33	80 × 40 × 18
Sampling Frequency [Hz]	10	5
Maximum Design Load/Full-Scale (F.S.) Output [kg]	90	150
Manufacturer Reported Accuracy	±(0.635 kg + 5% of the applied force)	Precision: 0.1% F.S. Nonlinearity: 0.1% F.S. Repeatability: 0.1% F.S. Hysteresis: 0.1% F.S
Smartphone App	Android/iOS	Android/iOS
Device Firmware	Version 1.0	Version 0.2.17
Cost [$USD]	149.99	135.00

**Table 2 sensors-21-08399-t002:** Intraclass correlations for test–retest reliability.

	Test Type		ICC(3,1) (95% CI)	F Test with True Value 0			
				Value	*df*1	*df*2	Sig
Activbody Activ5	Stepped Loads	Single measures	1.000 (1.000–1.000)	461,902.723	1355	2710	0.000
	Stress Relaxation	Single measures	1.000 (1.000–1.000)	120,719.759	503	1006	0.000
	Simulated Exercise	Single measures	1.000 (1.000–1.000)	24,656.891	421	842	0.000
	Hysteresis	Single measures	0.999 (0.999–0.999)	4060.913	837	1674	0.000
Tindeq Progressor	Stepped Loads	Single measures	1.000 (1.000–1.000)	26,976.380	702	1404	0.000
	Stress Relaxation	Single measures	0.999 (0.999–0.999)	6021.595	257	514	0.000
	Simulated Exercise	Single measures	0.999 (0.999–0.999)	3471.104	206	412	0.000
	Hysteresis	Single measures	1.000 (1.000–1.000)	43,333.090	445	890	0.000

**Table 3 sensors-21-08399-t003:** Pearson’s correlations, interclass correlations, and concordance correlations comparing both the Activ5 and Progressor with the gold-standard measure (Instron) across all conditions.

	Test Type	Pearson’s ρ (95% CI)		ICC(3,1) (95% CI)	F Test with True Value 0				ρ_c_ (95% CI)	C_b_
					Value	*df*1	*df*2	Sig		
Activbody Activ5 vs. Instron	Stepped Loads	0.999 (0.999–1.000)	Single measures	0.982 (0.944–0.991)	160.827	1289	1289	0.000	0.982 (0.981–0.983)	0.983
	Stress Relaxation	0.999 (0.999–1.000)	Single measures	0.971 (0.845–0.989)	122.599	439	439	0.000	0.971 (0.968–0.974)	0.972
	Simulated Exercise	1.000 (1.000–1.000)	Single measures	0.991 (0.941–0.997)	406.043	359	359	0.000	0.991 (0.990–0.992)	0.991
	Hysteresis	0.998 (0.998–1.000)	Single measures	0.990 (0.976–0.994)	260.336	789	789	0.000	0.990 (0.989–0.991)	0.992
Tindeq Progressor vs. Instron	Stepped Loads	1.000 (1.000–1.000)	Single measures	1.000 (1.000–1.000)	16,764.771	698	698	0.000	1.000 (1.000–1.000)	1.000
	Stress Relaxation	1.000 (1.000–1.000)	Single measures	1.000 (1.000–1.000)	17,035.523	219	219	0.000	1.000 (1.000–1.000)	1.000
	Simulated Exercise	0.999 (0.999–1.000)	Single measures	0.999 (0.999–1.000)	3086.294	179	179	<0.001	0.999 (0.999–0.999)	1.000
	Hysteresis	1.000 (1.000–1.000)	Single measures	1.000 (1.000–1.000)	116,988.081	424	424	0.000	1.000 (1.000–1.000)	1.000

ρ, Pearson’s correlation (precision). ρ_c_, Lin’s concordance correlation. Cb = ρ_c_/ρ, bias correction factor (trueness).

## Data Availability

Not applicable.

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
