# Peer review of "The Validity and Reliability of Two Commercially Available Load Sensors for Clinical Strength Assessment"

_sensors, 2021, doi:10.3390/s21248399_

Round 1

Reviewer 1 Report

The author use a universal mechanical testing machine to test two products in the market. It appears to be more like product reviews instead of developing new sensor component, sensing mechanism, algorithm. If proposed as a new medical device testing need to be included in medical/disease condition assessed by healthcare professionals with patients/human objects. 

Reviewer 2 Report

In general, the paper is well written and structured. English is fine. The aim is clear and useful for consumers. The experimental design is robust and the analysis of the results is well conducted.

title: I'd replace "load sensors" with "hand-held dynamometers" in the title to be more straight.

line 16: I wonder how an isometric dinamometer may provide mechanical power if there is no displacement. 

line 51-52: this assertion is only partially true as weight-resistance (i.e., isoinertial) muscle strength testing may be also used (cfr. Bosco C. Methods of functional testing during rehabilitation exercises. In: Puddu G, editor. Rehabilitation of sports injuries: current concepts. Berlin: Springer Ed; 2001. p. 11-22), for instance, for designing and monitoring a rehabilitation program in the injured shoulder (cfr., Tranquilli C, Bernetti A, Picerno P (2013). Ambulatory joint mobility and muscle strength assessment during rehabilitation using a single wearable inertial sensor. MEDICINA DELLO SPORT, 66(4):683-97). From an instrumental point of view, muscle strength testing through progressive weight-resistance can be either performed by means of linear encoders or accelerometers.

line 63: this assertion is not true. The fact that ID is a common practice (even if very expensive and, thus, not really ambulatory as only very specialized centers have this instrumentation because of its high cost) does not mean that it is a gold standard for measuring muscle strength in general. In fact, while isokinetic apparatus have the great advantage of avoiding tendon overloads by accommodating muscle tension variations over the whole arc of movement the muscle activation they produce is different from those characterizing typical human movements, which are not performed with fxed joint angles at constant joint angular velocities (Tranquilli C, Bernetti A, Picerno P (2013). Ambulatory joint mobility and muscle strength assessment during rehabilitation using a single wearable inertial sensor. MEDICINA DELLO SPORT, 66(4):683-97).

line 64: a bracket is probably missing for reference no. 18.

line 104: please specify here that the Instron ElectroPuls E10000 was considered as the gold standard.

line 125-126: please specify how the thermoplastic interface was shaped to fit the Active5 curvilinear profile.

line 136-137: please specify, if needed, also the charge of the batteries of the two devices (did you check whether the calibration keeps linear across different percentages of charge?)

Figure 2: why did you use different amplitudes of applied force between the two devices with the UMT?

line 180-188: why didn't you just use a simple resampling method like linear interpolation, cubic interpolation or spline?

line 212: a common test for assessing reliability (absolute agreement) between two methods (where one of which is the gold standard) is the Bland-Altman analysis. Why didn't you chose it?

line 444-446: please remove these lines.

Reviewer 3 Report

This paper assessed the validity and reliability of two commercially available Bluetooth load sensors (Activ5 by Activbody and Progressor by Tindeq) since there are no previous studies which have assessed the validity and reliability of the Activ5 or the Progressor for force measurement. The tests done to evaluate the 2 sensors were appropriate to validate the reliability of the sensors. Results are overwhelming although one sensor, the Activ5, performs worse than the gold standard. The following are my suggestions:

  1. The research findings presented in this paper are outstanding and is suitable to be published. However, there are many grammatical mistakes such as the citation formatting, grammar, etc. It is best for the authors to proofread the paper to an English language expert or through double-checking by the authors themselves.
  2. Figures are not clear. Please use high-definition images.
  3. If possible, writing should be done using Latex for easy formatting.
  4. In Figure 4, please separate the graphs and tables.
  5. In line 444, please remove:
    "This section is not mandatory but can be added to the manuscript if the discussion is unusually long or complex." Please double check the paper before submitting it to this journal.
